# Bio-Organically Acidified Product-Mediated Improvements in Phosphorus Fertilizer Utilization, Uptake and Yielding of *Zea mays* in Calcareous Soil

**DOI:** 10.3390/plants12173072

**Published:** 2023-08-27

**Authors:** Khuram Shehzad Khan, Muhammad Naveed, Muhammad Farhan Qadir, Muhammad Yaseen, Manzer H. Siddiqui

**Affiliations:** 1Institute of Soil & Environmental Sciences, University of Agriculture Faisalabad, Faisalabad 38000, Pakistan; khurramshehzadkhanniazi@gmail.com (K.S.K.); farhanqadir65@gmail.com (M.F.Q.); dr.yaseen@gmail.com (M.Y.); 2College of Resources and Environmental Sciences, National Academy of Agriculture Green Development, China Agricultural University, Beijing 100193, China; 3Department of Botany and Microbiology, College of Science, King Saud University, Riyadh 11451, Saudi Arabia; mhsiddiqui@ksu.edu.sa

**Keywords:** bio-acidulation, animal manure, P-solubilizing bacteria, phosphorus fertilizers, maize, microbial soil conditioner

## Abstract

The demand for a better agricultural productivity and the available phosphorus (P) limitation in plants are prevailing worldwide. Poor P availability due to the high pH and calcareous nature of soils leads to a lower P fertilizer use efficiency of 10–25% in Pakistan. Among different technologies, the use of biologically acidified amendments could be a potential strategy to promote soil P availability and fertilizer use efficiency (FUE) in alkaline calcareous soils. However, this study hypothesized that an acidified amendment could lower soil pH and solubilize the insoluble soil P that plants can potentially uptake and use to improve their growth and development. For this purpose, the test plant *Zea mays* was planted in greenhouse pots with a recommended dose rate of 168 kg ha^−1^ of P for selected phosphatic fertilizers, viz., DAP (diammonium phosphate), SSP (single superphosphate), and RP (rock phosphate) with or without 2% of the acidified product and a phosphorus solubilizing *Bacillus* sp. MN54. The results showed that the integration of acidified amendments and PSB strain MN54 with P fertilizers improved P fertilizer use efficiency (FUE), growth, yield, and P uptake of *Zea mays* as compared to sole application of P fertilizers. Overall, organic material along with DAP significantly improved plant physiological-, biochemical-, and nutrition-related attributes over the sole application of DAP. Interestingly, the co-application of RP with the acidified product and MN54 showed a higher response than the sole application of DAP and SSP. However, based on our study findings, we concluded that using RP with organic amendments was a more economically and environmentally friendly approach compared to the most expensive DAP fertilizer. Taken together, the current study suggests that the use of this innovative new strategy could have the potential to improve FUE and soil P availability via pH manipulation, resulting in an improved crop productivity and quality/food security.

## 1. Introduction

Phosphorus (P) is an important macronutrient for plant metabolic functions and growth [1], but its limited availability to plants has become a radical concern for crop production under different soils [2]. Many agricultural soils have both organic and inorganic P reserves, but the available P for plant uptake is quite low, at about <1 ppm [3]. In surface soil, the overall P contents range between 0.02 and 0.15%, and plant demand varies between approximately 0.2 and 0.8% [4]. Moreover, about 5.7 billion hectares of agricultural area is P-deficient, which ultimately causes 30–40% yield losses worldwide [5,6]. Therefore, systematic and sustainable P management practices are needed to maintain soil fertility, productivity, and global food security [7].

Similarly, P deficiency is a limiting factor that causes stunted plant growths and, ultimately, severe losses in crop productivity [8]. In Pakistan, over 90% of the soil’s fertility rate is reducing due to higher pH (>7.5) and CaCO_3_ (>3%). The higher calcium ion concentration in calcareous soils results in a lower P availability or P-fixation with Ca and Mg ions. Similarly, P in red acidic soils also develops a very strong bond with Al and Fe ions [9]. The large area of about 80 to 90% of Pakistani sub-soils is known as P-limited soils. A relatively small amount of applied phosphatic fertilizers (about 5–20%) are utilized by crops and a large portion becomes unavailable in soil [10], which result in very low P utilization. The different inputs of P nutrients are applied to the soil, either chemical P fertilizers or organically enriched P amendments [11]. The major chemical fertilizer (diammonium phosphate, DAP) used in Pakistan is a costly fertilizer for the farming community [12]. Its use efficiency is very low, at about 20%; therefore, it is an urgent need to develop innovative and sustainable technologies for promoting PUE. Moreover, agricultural yields with less dependency on chemical fertilizers has brought our immediate attention to the benefits of the farming community [13,14].

Rock phosphate (RP) is a naturally occurring mineral source of phosphorus (P). In Pakistan, there are around 6.9 million tons of rock phosphate reserves [15]. It is a less expensive and more efficient P natural resource as compared to synthetic fertilizers, mainly DAP. In contrast, the major drawback of RP use as phosphatic fertilizer is its low solubility, but it can be fixed by using some sustainable approaches to improve PUE [16]. These approaches include a partial RP acidulation with organic-based materials, which might increase the solubility of rock phosphate [17,18]. Organic manures or residues, through improvements in soil physicochemical characteristics and the efficient uptake of mineral nutrients, promote crop biomass and yields [19,20,21]. Animal dung has been reported widely to cause partial acidulation, the solubilization of RP, as well as improve soil health and properties [22]. Previous studies exhibited that P solubility is mostly associated with soil pH, where a higher P availability at soil pH 6–7 is reported [7,23]. The challenge of the bulk dairy manure management could be overcome through sustainable mixing of nature-based organic manures with RP into organic alternative P sources. Furthermore, a composting approach using enriched RP manures could convert these limited reserves into more soluble forms that are less expensive and are more efficient in their natural resource use efficiency [24]. Although the composting rate can be enhanced through P-solubilizing bacteria (PSB) [25], consequently, bacteria-assisted compost requires less time to make an applicable form of composted RP in soil [26]. On the other hand, different types of microbes (PGPR) produced organic acids that provide a more available form of P [27,28]. Similarly, Nadeem et al. [29] suggested that sulfur amended farmyard manure, and when augmented with sulfur-oxidizing bacteria (SOB), produced acidified compost (pH < 2.00), which significantly improved the wheat growth and P availability cultivated under calcareous soils. The absence of soluble carbon and nitrogen sources results in restricted PSB activities, microbial biomass, and the P solubilization process. Thus, the stimulation of PSB through organic C and N sources not only increases the decomposition of complex material structures, but also increases the solubilization rates of phosphatic rocks [30,31]. 

Since this study, different techniques such as organic, chemical fertilizers, and microbial inoculants have been applied to increase PUE under different soil and climate [32,33]. Taking this background into account, the current development of novel technology in sustainability includes the usage of bioaugmented (PSB and SOB) and acidulated (sulfur-enriched) organic manure products to improve fertilizer use efficiency (FUE) and plant P nutrition, through the regulation of the root microenvironment by oxidizing sulfur and lowering soil pH, thus enhancing nutrients’ cycling and availability. Therefore, our research study hypothesized that the use of acidulated organic manure may improve the availability of P for maize plants, thereby promoting plant biomass, seed nutritional quality, and maize yield in calcareous soils. The core objective of the current study was to evaluate the potential of acidulated manure in combination with the PSB strain to improve fertilizer use efficiency (FUE) and soil PUE for a better plant growth and maize yield.

## 2. Materials and Methods

### 2.1. Acidulated Organic Manure Preparation and Analysis

The bioaugmented acidulated organic product used in this study was collected from the Soil and Environmental Microbiology Lab 31°26′ N 73°04′ E, University of Agriculture Faisalabad (UAF), Pakistan. Briefly, acidulated organic manure (AOM) was prepared by mixing elemental sulfur and cow manure bioaugmented with sulfur-oxidizing bacteria (SOB). The SOB, including *Thiobacillus* sp., was collected from the Soil and Environmental Microbiology Laboratory. The AOM was pre-analyzed for pH 1.8 ± 0.15 (1:2.5 *w*/*v*), C 36.9%, N 2.87%, P 1.25%, K 1.95%, Cu 27.9 mg kg^−1^, Zn 121.5 mg kg^−1^, and Mn 48.45 mg kg^−1^, respectively. Afterward, the acidified extract (liquid-acidulated organic manure, LAOM) was also prepared by extraction at a 1:2.5 ratio (*w/v*) with good quality irrigation water. It was prepared before the start of the experiment.

### 2.2. Preparation of Plant Growth Promoting Bacterial Inoculum and Soil Microbial Count Evaluation

A pre-isolated plant-growth-promoting (PGP) bacterial strain, *Bacillus* sp. MN54 [33], was collected from the Soil and Environmental Science Lab, Institute of Soil and Environmental Sciences (ISES), UAF, Pakistan. This strain was grown separately in LB (Luria–Bertani) broth media, containing 10 g L^−1^ tryptone, 5 g L^−1^ yeast extract, and 10 g L^−1^ NaCl at 28 ± 1 °C, and 100 rev min^−1^ for 48 h in an orbital shaking incubator (Firstek Scientific, Japan). To attain a uniform bacterial growth of 10^8^–10^9^ CFU ml^−1^ (colony forming units), the spectrophotometer at 600 nm (Gene Quant Pro, Gemini B.V., Apeldoorn, The Netherlands) was used. Furthermore, the soil microbial count of the pot study was analyzed by using a serial dilution plating technique on tryptic soy broth (TSB) and the number of viable cells were estimated as colony-forming units (CFU g^−1^ soil) as described by [34].

### 2.3. Pot Study and Detail

A greenhouse pot study was set up at the research station of the ISES 31.7° N, 73.98° E, UAF, Pakistan. Soil was sandy clay loam (sand 45%, silt 28%, and clay 24%) and a soil saturation percentage of 32% was taken from field plots of the ISES research station, air-dried, sieved (<2 mm), and analyzed for physical and chemical properties. The initial soil properties were pH 8.2 (5:1; water/soil), CaCO_3_ 27.8%, Ca + Mg 15.3 me L^−1^, CO_3_ 1.98 me L^−1^, HCO_3_ 3.6 me L^−1^, Cl 19.8 me L^−1^, SO_4_ 30.2 me L^−1^, Olsen-P 6.58 mg kg^−1^, 0.48 g total P kg^−1^, organic matter 0.69%, cation exchange capacity (CEC) 12.9%, 115.5 available K mg kg^−1^, and total N 0.086%. These basic characteristics of soil were calculated by the following methods: carbonate and soluble salt through volumetric titration method; the SO_4_ turbidimetric method were proposed by the US Salinity Laboratory Staff [35]; total N was estimated using Jackson’s method [36]; total P was obtained by following the method of Ryan et al. [37]; available K content [38] and (CEC) were measured by using Rhoades’s [39] method; and organic matter was measured using Moodie’s method [40]. The experiment treatments consisted of three P fertilizer sources (rock P, RP; single superphosphate, SSP; diammonium phosphate, DAP) under 2% of acidulated manure (solid and liquid forms) along with and without PSB strain (MN54). In control condition (CK), there was no P added. Before the start of the pot experiment, all treatments were uniformly mixed with soil. The P fertilizer sources were applied at the recommended rate of P 168 kg ha^−1^. The bacterial broth of 50 mL per pot was also inoculated before sowing time and fertigated twice at the growth and reproductive stages of maize. Additionally, nitrogen (urea) and potassium (MOP, muriate of potash) were applied in splits with the recommended dose of 181:131 kg ha^−1^ (N:K) for normal plant growth and to avoid nutrient deficiency [41]. Each treatment was replicated three times with a completely randomized design.

### 2.4. Test Plant, Harvest, and Sample Analysis

Maize seeds (P4040) were provided by ISES. In each pot, six maize seeds (surface sterilized with 70% ethanol for 2 min and 1.5% NaClO for 5 min, followed by thrice washing with autoclaved distilled water) were sown. Each pot was 24 cm in height and 32 cm in diameter and contained 8 kg of soil. After seedling, plants were thinned to ensure one plant for each pot. During the experiment, the good quality canal water was supplied daily to adjust the soil moisture content to 18% (*w/w*). The pots were rearranged every week by following the completely randomized design. The plants were harvested after 123 days and further processed for growth, yield, and physiological analysis. 

After harvesting, three primary soil samples randomly collected from each pot were mixed into one composite sample, which was then sieved (<2 mm), air-dried, and stored for chemical analysis. Soil Olsen P was measured by 0.5 M NaHCO_3_ extraction (pH = 8.5), following the Olsen method [42], and soil organic carbon (SOC) was measured through oven-dried and sieved (0.25 mm) soil using the Walkley and Black method [43].

### 2.5. Plant Physiology and Nutrition Status 

The physiological traits of the maize plant were analyzed at midday (during 10:00–14:00). A multi-gas analyzer IRGA (CI-340), Germany, was used to measure stomatal conductance, assimilation rate, transpiration rate, water use efficiency, and chlorophyll contents at the vegetative stage of the fully mature plant; while the chlorophyll was recorded of the third leaf from apex with the help of chlorophyll meter (SPAD-502). Before drying the maize cob (yield-related attribute), other cob-related indexes such as cob dry weight, diameter, and length were recorded. Similarly, maize cobs and grains were sun-dried to present a constant weight. At the time of soil sampling, plant samples (above and below ground) were collected separately. The plant samples (root, shoot, and grain) were obtained after drying at 65 ± 1 °C for 72 h, and threshed for wet digestion using the method of Wolf [44]. After sample digestion, the P concentration was analyzed by following the colorimetric method of Bhargava and Raghupathi [45]. Further, the maize grains were also used for the determination of the quality parameters of crude fat, fiber, protein, and ash content [46]. The plant and soil P use efficiency indexes were determined by following the formula given below [41]:Total P-uptake (%) = P-uptake (Grain%) + P-uptake (Shoot%) + P-uptake (Root%)
and
P−uptake (mg/kg)={biomass (Oven dry)×P(%)}/100
$$\mathrm{PUE}\left(\%\right)=\frac{\mathrm{Total}\;P\mathrm{uptake}\mathrm{by}\mathrm{fertilized}\mathrm{plant}-\mathrm{Total}P\mathrm{uptake}\mathrm{by}\mathrm{unfertilized}\mathrm{plant}}{\mathrm{Amount}\mathrm{of}\mathrm{fertilizer}\mathrm{applied}}\times 100$$
where PUE is phosphorus use efficiency (%).

### 2.6. Statistical Analysis

Study results were analyzed using the software STATISTIX version 8.1. To analyze the effect of treatments on documented data, three-way ANOVA (analysis of variance) was used. Tuckey test was used to evaluate the significant difference between treatments (<0.05). The Pearson correlation between growth-related parameters was performed in RStudio, and a correlogram was constructed using the built-in “cor” function and the publicly available package “corrplot”, as per R-project instructions (RStudio Team, Boston, MA, USA, 2019). Origin Pro 2021 software was used to observe the positive relation with their treatments under principal components analysis (PCA).

## 3. Results

The mean values of the plant’s growth, yield, and biochemical traits of maize shoot, grains, and post-harvest soil analysis were recorded. A synergistic interaction with acidified organic manure was noticed, i.e., both liquid and solid applied with different phosphate fertilizers (RP, SSP, DAP), in the presence of plant growth-promoting endophyte (Bacillus MN54), which have the highest phosphorus solubilization characteristic. The results showed that maize traits were significantly influenced by a 2% liquid-acidified organic manure (LAOM) or solid-acidified organic manure (AOM) with RP. As P utilization was maximum with this biodynamic approach because bio-organically acidified product efficiently solubilized RP and P became available to plants, and a similar trend was also observed for DAP through the addition of LAOM or AOM to maize. The application of RP therefore reduced the pressure at DAP and RP is more economical alternative.

### 3.1. Effect of LAOM and AOM with Different P sources along PSB on Maize Growth Attributes

The minimum value for plant height was recorded at control RP, as shown in Table 1. An increasing trend in plant height was observed with P sources (SSP, RP, and DAP) with a combination of acidified organic amendments (LAOM and AOM) along PSB application. In the case of LAOM, the application of RP with PSB (+) maximum increase (200%) was recorded in response to the respective control (CK RP) corresponding to the DAP (132.5%) and SSP (129%) with respect to their control treatments (CK DAP and SSP), respectively. Similarly, an increasing trend in the height of plants was detected with the amendment of AOM and phosphate fertilizers (SSP < RP < DAP), and the significant increase (208%) in height was noted at the combination of RP along with the PSB, as shown in Table 1. For the leaf area, a consistent increasing trend was investigated. With the input of LAOM, the combined use of RP and PSB showed the highest increase (185.5%) as compared to CK RP, while the highest leaf area was noticed with the integrated use of LAOM, DAP, and PSB as shown in Table 1. The overall application of AOM with P sources (SSP, RP, and DAP) showed a significant increase in leaf area, while the maximum increase (189%) as compared to CK RP was revealed via the synergistic application of AOM with RP and PSB (Table 1, Figure 1).

Through the application of both LAOM and AOM, an increasing trend in root length, shoot, and root dry weight was recorded with phosphate fertilizer in following order (DAP > RP > SSP) with combination of PSB. The highest increase (188.8%) in root length was recorded with the cumulative application of LAOM, RP, and PSB, followed by respective treatment (CK RP). Similarly, the maximum increase in shoot dry weight (206.7%) through the integrated application of LAOM with RP and PSB was observed, while the same treatment also showed the highest increase (239%) in root dry weight than that of the respective control (CK RP), as shown in Table 1.

The application of both LAOM and AOM showed a positive interaction between the maize yield attribute through different phosphate fertilizers with PSB; the combined application of LAOM, RP, and PSB showed the highest increase (230.5%) with respect to (CK RP) treatment under the co-application of AOM, RP, and PSB (198.8%) with respect to CK RP. While the LAOM and AOM significantly increased the fertilizer use efficiency via RP > DAP > SSP, the same trend was observed in the cob dry weight as shown in Table 1. The combined application of LAOM and RP with PSB showed the highest increase (182%) in cob length with respect to the control (CK RP), while this cob length was also high (121.2%) as compared to the common farmer practice of DAP control (CK DAP); moreover, with the application of AOM, RP, and PSB, the cob length also significantly improved (179%). Although the application of LAOM in combination with DAP and PSB showed the highest cob length with respect to all other treatments of RP and SSP, and cob length was increased by 125% with respect to the respective control (CK DAP) as shown in Table 1. Additionally, the combined application of LAOM, PSB, and RP showed a higher increase (251%) in cob diameter than (CK RP); furthermore, AOM also increased (228.5%) with the same combination of RP and PSB as shown in Table 1.

### 3.2. Effect of LAOM and AOM with Different P Sources along PSB on Maize Physiological and Nutritional Attributes

The application of both LAOM or AOM in combination with phosphate fertilizers (RP, SSP, DAP) with or without PSB significantly improved the physiological attributes of maize. The highest increase in chlorophyll contents (172.1%), stomatal conductance (233.4%), photosynthetic rate (233.1%), transpiration rate (302.2%), and water use efficiency (137.4%) was recorded with the co-application of LAOM, RP, and PSB when compared with the control (CK RP). Similarly, an increasing trend was noticed with the application of LAOM with SSP and DAP; the highest value of physiological attributes was reported with the acidified organic manures and DAP along with the PSB (Table 2).

The chemical analysis of plant biomass (shoot, root, and grains) showed a positive correlation toward LAOM or AOM application with different phosphate fertilizers (RP > DAP > SSP) with or without PSB. The maximum shoot phosphorus contents was measured under the combined use of LAOM, RP, and PSB (248.1%) as compared to the respective control (CK RP). Similarly, the chemical analysis of the plant’s roots and grains also showed that the co-application of LAOM, RP, and PSB increases P to reach its highest content in roots (258.9%) and in grains (218.1%) as compared to the respective control (CK RP), as shown (Table 2). 

### 3.3. Effect of LAOM and AOM with Different P Sources along PSB on Maize Seed Quality Traits

The combined application of LAOM or AOM with phosphate fertilizer in the presence of PSB significantly improved the maize grain quality and yield parameters (Table 3). An increasing trend in ash contents was recorded by the application of LAOM with the combination of following order DAP > RP > SSP along with PSB; and the maximum increased (205.4%) as compared to the control (CK RP). Furthermore, in terms of fat in grain for both experiments, the highest increase (216.3%) was recorded in the LAOM, RP, and PSB as compared to the respective control (CK RP). Similar findings were observed in the case of crude fiber and protein contents in grains with the maximum values at LAOM, RP, and PSB, at 233% and 150.9%, respectively, in comparison to control (CK RP), as shown in Table 3. The maximum thousand grain weight was reported with co-application of LAOM, DAP, and PSB, while the highest increment in thousand grain mass was calculated with respect to the sole application of phosphate fertilizer. The combined application of AOM, PSB, and RP, increases thousand grains weight (209.5%) with respect to the control (CK RP) followed by LAOM, PSB, and RP (205%), as shown in Table 3.

The three-way ANOVA of all these parameters of Table 1, Table 2 and Table 3 is presented in Appendix A, where there is a significant effect of bacterial inoculation and acidified organic amendments, and the forms of fertilizers could be evaluated.

### 3.4. Effect of LAOM and AOM with Different P Sources along PSB on Post-Harvest Soil Analysis

The mean values of post-harvest soil analysis revealed that the co-application of LAOM or AOM with phosphate fertilizers and PSB had a significant impact on the selected attributes of experimental soil, as shown in Figure 2. A five-fold increase in phosphorus uptake was recorded, where a combination of LAOM, RP, and PSB was applied, as compared to the respective control (CK RP), and the maximum (546.3 mg kg^−1^) significant value was recorded at the combined application of LAOM, DAP, and PSB (Figure 2A). Alternatively, in the case of soil extractable P, an increasing trend was found in the combination of LAOM or AOM with phosphate fertilizer in the following order as DAP > RP > SSP along with PSB. The highest increase in soil extractable P (194.4%) was monitored, where the co-application of LAOM, RP, and PSB was applied as compared to the specific control (CK RP) (Figure 2B). A small decrease in pH was recorded where LAOM or AOM with P-solubilizing bacteria was used (Figure 2C). The phosphorus use efficiency (PUE) significantly improved with the application of acidified manure, while the highest increase of 115% was seen in combination with the application of LAOM, RP, and PSB as compared to AOM, RP, and PSB (Figure 2D).

An increasing trend in microbial count and soil organic carbon was reported with the combined application of acidified manures, PSB, and different phosphate fertilizers. The highest (84 × 10^7^) increase in bacterial count was seen with the combined application of LAOM, PSBs and DAP, followed by SSP and RP (Figure 3A). Soil organic carbon (SOC) was fluctuating with the co-application of acidified organic amendments with PSB and phosphate fertilizers (RP > DAP > SSP), and the maximum SOC was observed with the AOM, RP, and PSB application as compared to the respective counterpart of LAOM, RP, and PSB (Figure 3B).

### 3.5. Relationship between Maize Growth Parameters under Different Treatments

The Pearson correlation depicted the existence of a strong correlation between maize growth attributes (Figure 4). Root length (RL) was positively correlated to cob length (CL) (r = 0.97), cob diameter (CD) (r = 0.94), plant height (PH) (r = 0.85), root dry weight (r = 0.95), and cob dry weight (CDW) (r = 0.84). Plant height was positively correlated with thousand grain weight (KGW) (r = 0.90). In addition, chlorophyll content (CHL) exhibited a strong correlation with stomatal conductance (STC) (r = 0.92), transpiration rate (TR) (r = 0.85), and photosynthetic rate (PSR) (r = 0.82) (Figure 4).

The PCA analysis showed that PSBAOMRP (PSB+, 2% acidified amendment, and rock P), PSBAOMSSP (PSB+, 2% acidified amendment, and SSP), PSBAOMDAP (PSB+, 2% acidified amendment, and DAP), PSBLAOMRP (PSB+, 2% liquid-acidified amendment, and rock P), and LAOMDAP (2% liquid-acidified amendment, and DAP) are positively correlated with most of the parameters used in this study as shown in Figure 5. The important parameters include P in soil (SOILP), crude fat (CFAT), crude fiber (CFIB), crude protein (CPRO), ash contents (ASHC), grain P (GRP), and P use efficiency (PUP). When preceding parameters were considered, PSBAOMRP (PSB+, 2% acidified amendment, and rock P), PSBAOMSSP (PSB+, 2% acidified amendment, and SSP), and PSBAOMDAP (PSB+, 2% acidified amendment, and DAP) outperformed the remaining treatments (Figure 5).

## 4. Discussion

Phosphorus (P) limitation in high pH calcareous soils is a serious issue for better yield and crop production. In Pakistan, P deficiency and its availability is hindered by fixation or precipitation, due to a higher CaCO_3_ content in soil. For combating this, strong chemical fertilizers have been used, while the elevated price of DAP is still a conundrum for farmers’ affordability and natural resource use efficiency. As an alternative, the use of RP is an appealing option due to its reasonable price, but its solubility in our soils is very low, which is a major hindrance to its direct use by farmers. Hence, to keep in view the whole scenario, an economical alternative approach of using bioaugmented acidified amendments to increase soil PUE and the solubility of RP can be enhanced as a dual benefit. 

The application of phosphatic fertilizers with acidified organic amendment and phosphate-solubilizing bacteria (PSB) had significantly (*p* < 0.05) enhanced the soil P bioavailability. Our findings are in parallel with [47], whose authors demonstrated that soil P availability could be improved by solubilizing fixed or precipitated P using acidified amendments along PSB. In fact, the acidified amendment and acid-producing agents suddenly/immediately lower the soil pH as well as solubilize the fixed and precipitated P in calcareous soils [47]. This study supports the fact that minimum changes in pH could cause a major effect in soil nutrient availability, especially P, in calcareous soil where nutrient availability is dependent upon pH [48,49,50]. Our study results are consistent with Hashemimajd et al. [51], who stated that acidifying amendments significantly improved the P concentration and confirmed a strong correlation between soil acidity and P availability in alkaline soil [52]. Therefore, it could be an effective strategy to improve P fertilizer use efficiency and to manage soil P fertility in the calcareous soil.

In the current study, the co-application of acidified organic manure amendment along with PSB not only increased the soil organic carbon, but microbial biomass as well. This indicates that the addition of organic manures increases the below-ground C contents, which might increase soil microbial biomass [53]. Since the treated soil was significantly deficient in organic material (less than 0.7%), the input of a high C-containing source can cause changes in the availability of organic carbon in the soil, which consequently exerted a significant impact on the growth and microbial biomass by using C as an energy source [54,55]. 

Interestingly, we found that acidulated material had neutralized the impact on the basic higher soil pH, which in turn showed a positive relationship with soil P availability. Our findings are consistent with [56,57], whose authors reported that in the presence of certain amendments as the utilization of P fertilizers, especially RP, significantly improved microbial P solubilization in alkaline calcareous soil. Study results also indicated that the enhancement of soil P availability for efficient plant P uptake might be due to the presence of PGPR. In fact, PSB has the potential to make P in a more bioavailable form for uptake by plants [58]. However, it has been suggested that organic manures bioaugmented with PSB has great potential to make a more available form of P for plant uptake [59]. 

Study results manifested that the addition of acidified amendment along P fertilizers improved not only FUE and PUE, but also enhanced the growth and yield of maize as well. The leading combination of acidified manure and DAP effectively promoted yield-related factors, viz., above- and below-ground plant biomass over sole SSP, DAP, or RP. Comparatively, plant physiological parameters, i.e., chlorophyll content, showed prominent results when SSP, DAP, and RP were applied alone, which might have been because of the failure of these phosphatic fertilizers to facilitate available nutrients, particularly P for plant uptake. Likewise, the soluble P forms (SSP and DAP) were exposed to fixation with soil particles, so that at critical stages, they might not present an adequate quantity of P due to P accumulation either through chemical adsorption or precipitation, which might have decreased the maize yield [60]; similarly the photosynthetic rate, stomatal conductance, and assimilation rate were also increased by acidified manure treatments in combination with DAP. These variable conditions might have increased the organic matter and nutrient contents, soil porosity, water-holding capacity, and, specifically, soil pH reduction, which provided uninterrupted nutrient availability to meet plant nutritional demands, resulting in a better yield quality [61,62].

In keeping the above discussion in view, the RP with acidulated manure showed either a positive influence on plant growth and P utilization by crop, followed by SSP, or almost equal results with only DAP application. Therefore, it is indicated that acidulated amendment could potentially solubilize the RP for mineral nutrient (such as P) uptake by crops, ultimately helping to improve phosphatic fertilizer utilization efficiency. Thus, an integrated use of novel organic manure could be an effective and new economical approach to boost the soil biological status, improve fertilizer use efficiency, and resolve the problem of calcareousness to achieve fertile soil for sustainable crop production.

## 5. Conclusions

It was concluded that physiological attributes, nutrient uptake, and the seed quality parameters of maize were improved through an integrated approach of regulating acidified organic manure with different phosphatic fertilizers and PSB via a combined application. The current study suggested that the use of this new innovative strategy could have the promising potential to improve FUE and soil P availability via soil pH manipulation and, as a result, improve crop productivity. More importantly, the utilization potential of rock phosphate resources could cost-effectively ameliorate the issue of the extensive cost required to import phosphatic fertilizers, i.e., DAP from other countries to improve maize yield, both quantity- and quality-wise. Taken together, this effective and alternative approach further needs to be explored in broad-acre farming systems to achieve the goal of the minimum use of chemical fertilizers for sustainable agriculture production in an economical and environmentally friendly way. 

## Figures and Tables

**Figure 1 plants-12-03072-f001:**
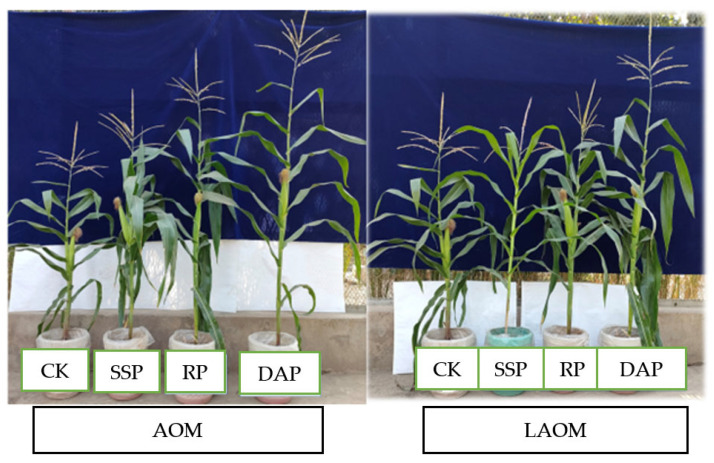
Depiction of plant growth under the AOM and LAOM with P fertilizers along PSB. Where CK means control without acidified amendment and PSB; SSP, single superphosphate; RP, rock phosphate; DAP, diammonium phosphate.

**Figure 2 plants-12-03072-f002:**
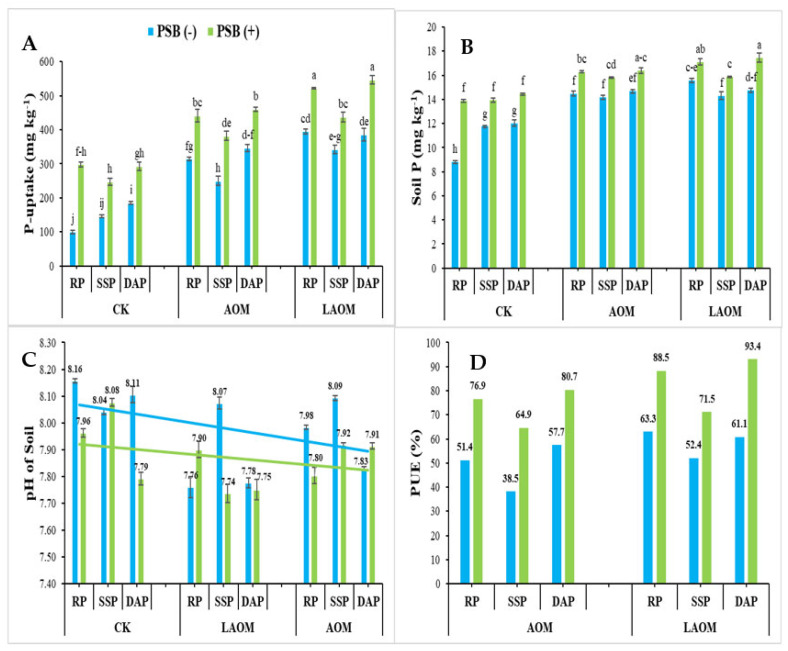
Effect of LAOM (liquid-acidified organic amendment) and AOM (acidified organic amendment) with different P sources along PSB on (**A**) P uptake (mg/kg), (**B**) soil available P (mg/kg), (**C**) soil pH, and (**D**) PUE (%). Where treatments of CK represents the control condition with no P added; RP, rock phosphate; SSP, single superphosphate; DAP, diammonium phosphate. Different letters along with figures indicate the significant differences (*p* < 0.05) of PSB and acidified organic matter to P fertilizers.

**Figure 3 plants-12-03072-f003:**
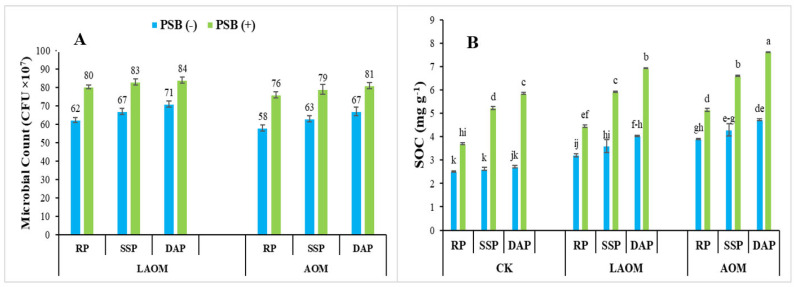
Effect of LAOM and AOM with different P sources along PSB on (**A**) microbial bacterial count (CFU × 10^7^) and (**B**) SOC: soil organic C (mg g^−1^). Where treatments of CK represent control condition with no P added and nor acidulant product was added; RP, rock phosphate; SSP, single superphosphate; DAP, diammonium phosphate. Different letters along with figures indicate the significant differences (*p* < 0.05) of PSB and acidified organic matter to P fertilizers.

**Figure 4 plants-12-03072-f004:**
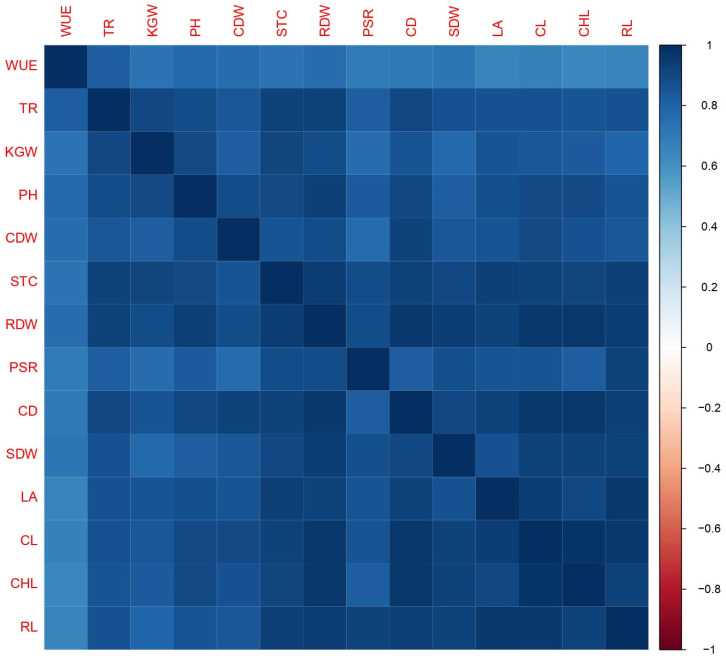
Correlation pattern (Pearson) of plant growth, physiological, biochemical, and yield index. WUE (water use efficiency), TR (transpiration rate), KGW (thousand grains weight), PH (plant height), CDW (cob dry weight), STC (stomatal conductance), RDW (root dry weight), PSR (photosynthetic rate), CD (cob diameter), SDW (shoot dry weight), LA (leaf area), CL (cob length), CHL (chlorophyl content), and RL (root length).

**Figure 5 plants-12-03072-f005:**
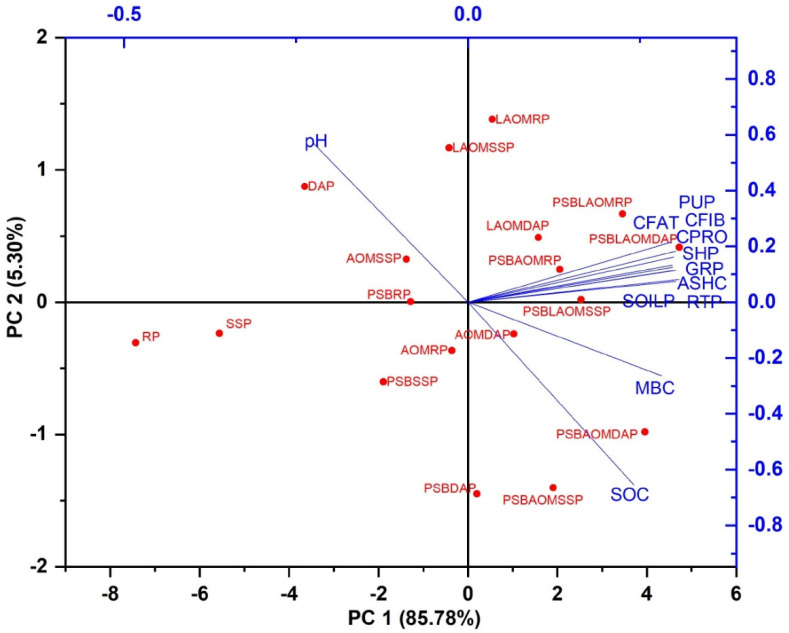
Principal component analysis (PCA) of plant growth, physiology, nutritional, gaseous exchange, and yield indices as well as soil post-harvested index of maize grown. Biplot showing scores in the first two principal components (PC1, *x*-axis; PC2, *y*-axis) for traits, (blue lines are dependent variable and lie under positive matrix showing the correlation with respect to treatments as crude fat (CFAT), crude fiber (CFIB), crude protein (CPRO), ash content (ASHC), grain P (GRP), and P use efficiency (PUP), pH (soil pH), soil P (soil available P), MBC (microbial count)). In addition, red dots are the treatments, PSBAOMRP (PSB+, 2% acidified amendment, and rock P), PSBAOMSSP (PSB+, 2% acidified amendment, and SSP), and PSBAOMDAP (PSB+, 2% acidified amendment, and DAP).

**Table 1 plants-12-03072-t001:** Effect of LAOM and AOM with different P sources along PSB on maize growth parameters: plant height (PH), leaf area (LA), root length (RL), shoot dry weight (SDW), root dry weight (RDW), cob dry weight (CDW), cob length (CL), and cob diameter (CD). Different letters along with figures indicate the significant differences (*p* < 0.05) of PSB and acidified organic matter to P fertilizers.

Bacteria	O.M.	P Source	Parameters							
			PH (cm)	LA (cm^2^)	RL (cm)	SDW (g)	RDW (g)	CDW (g)	CL (cm)	CD (mm)
PSB (−)	CK	RP	90.5 ± 3.6 h	272 ± 6.05 g	56.83 ± 1.93 g	45.33 ± 1.21 j	12.8 ± 0.89 f	29.5 ± 1.05 f	11.16 ± 0.73 e	7.25 ± 0.27 f
		SSP	138.17 ± 4.6 g	279.17 ± 3.5 g	66.67 ± 0.91 g	59.33 ± 1.37 i	19.7 ± 0.7 e	40.5 ± 2.38 ef	14.3 ± 0.46 d	10.08 ± 0.37 e
		DAP	146.33 ± 2.3 g	439.5 ± 4.8 ef	90.3 ± 2.65 d–f	64.7 ± 1.2 hi	21 ± 1.53 de	47 ± 2.76 de	16.5 ± 0.61 cd	11.62 ± 0.32 e
	AOM	RP	163 ± 2.04 f	455.33 ± 1.7 df	90 ± 1.16 ef	69.3 ± 0.88 gh	25.3 ± 0.8 a–e	57 ± 1.73 b–d	19 ± 0.58 a–c	15.6 ± 0.46 cd
		SSP	177.3 ± 2.03 c–e	437.67 ± 1.4 ef	85.33 ± 1.86 f	63.33 ± 0.88 i	23 ± 0.6 c–e	57.7 ± 5.2 a–d	17.3 ± 0.67 bc	15 ± 0.58 d
		DAP	171 ± 2.3 d–f	439.33 ± 8.9 ef	97.7 ± 4.1 b–e	77 ± 1.16 d–f	26 ± 0.6 a–e	59 ± 3.61 a–d	18 ± 0.58 a–c	16.3 ± 0.4 b–d
	LAOM	RP	161.66 ± 1.4 f	470.7 ± 2.8 cd	102 ± 2.52 a–d	86.33 ± 0.58 b	26.7 ± 1.5 a–d	51 ± 1.77 c–e	19.3 ± 0.58 a–c	16.6 ± 0.4 a–d
		SSP	171.33 ± 2.3 d–f	455 ± 3.2 de	89.3 ± 2.41 ef	79 ± 1.73 c–e	25.7 ± 0.8 a–e	65.7 ± 2.9 ab	18.3 ± 0.33 a-–	15.7 ± 0.57 cd
		DAP	175.67 ± 1.7 c–e	464.6 ± 4.3 de	102.3 ± 3.5 a–c	82.7 ± 0.88 bc	26.7 ± 1.2 a–d	67 ± 2.08 ab	19.3 ± 0.33 a–c	16.7 ± 0.4 a–d
PSB (+)	CK	RP	139.5 ± 1.6 g	449.3 ± 3.6 de	96.3 ± 2.14 c–f	73 ± 0.87 fg	23.3 ± 1.4 b–e	57.1 ± 1.7 a–d	18.5 ± 0.58 a–c	15.6 ± 0.33 cd
		SSP	167 ± 2.3 ef	421 ± 5.8 f	87.5 ± 2.1 ef	69.5 ± 0.87 gh	23.3 ± 1.4 b–e	57.5 ± 2.4 a–d	17.1 ± 0.3 b–d	15.07 ± 0.38 d
		DAP	170.5 ± 2.4 d–f	461.2 ± 4.2 de	98.3 ± 1.9 b–e	73.8 ± 1.5 e–g	26.7 ± 0.6 a–d	61.7 ± 3.4 a–c	18.2 ± 0.61 a–c	16.4 ± 0.5 b–d
	AOM	RP	185.33 ± 1.4 a–c	510 ± 0.4 ab	105.3 ± 1.9 a–c	82 ± 1.16 b–d	29.7 ± 0.7 ab	58.7 ± 2 a–d	20 ± 0.58 ab	16.6 ± 0.23 a–c
		SSP	181.33 ± 2.4 b–d	493 ± 1.5 bc	98.7 ± 2.6 b–e	77.7 ± 1.45 c–f	26.7 ± 2.7 a–d	59.7 ± 4.7 a–d	19 ± 0.58 a–c	17.3 ± 0.33 a–c
		DAP	193.67 ± 1.4 ab	530 ± 8.6 a	109 ± 1.16 ab	83 ± 2.08 bc	30.33 ± 1.2 a	61.3 ± 4.9 a–c	20 ± 0.58 ab	18 ± 0.29 ab
	LAOM	RP	184.67 ± 2.3 a–c	510 ± 4.1 ab	107.3 ± 2.4 a–c	93.67 ± 0.88 a	30.7 ± 1.5 a	68 ± 1.53 ab	20.3 ± 0.88 a	18.2 ± 0.53 ab
		SSP	180.33 ± 1.4 cd	497.7 ± 1.8 bc	103.3 ± 1.5 a–c	85.33 ± 1.2 b	28.3 ± 2.6 a–c	68.7 ± 2 ab	20.3 ± 0.67 a	18 ± 0.29 ab
		DAP	196.67 ± 1.6 a	533 ± 4.1 a	111.3 ± 2.34 a	95.67 ± 1.2 a	31.3 ± 1.2 a	70 ± 3.06 a	20.7 ± 0.88 a	18.67 ± 0.33 a

**Table 2 plants-12-03072-t002:** Effect of LAOM and AOM with different P sources along PSB on maize physiology and nutrition index: thousand grain weight (KGW), chlorophyll content (CHL), stomatal conductance (STC), assimilation rate (ASR), transpiration rate (TRR), water use efficiency (WUE), P contents in shoot (shoot–P), P contents in root (root–P), and P content in grain (grain–P). Different letters along with figures indicate the significant differences (*p* < 0.05) of PSB and acidified organic matter to P fertilizers.

Bacteria	O.M.	P Source	CHL (mg g^−1^)	STC (mmol m^−2^ s^−1^)	ASR (µmol CO_2_ m^−2^ s^−1^)	TRR (mmol m^−2^ s^−1^)	WUE (%)	Shoot–P (%)	Root–P (%)	Grain–P (%)
PSB (−)	CK	RP	35.5 ± 1.17 e	38.83 ± 2.25 j	14.6 ± 1.16 d	1.29 ± 0.21 g	7.33 ± 0.67 d	0.18 ± 0 i	0.13 ± 0.01 h	0.22 ± 0.02 f
		SSP	44.33 ± 1.2 d	45.83 ± 1.31 j	22.9 ± 1.03 c	1.57 ± 0.25 fg	8.32 ± 0.24 a–d	0.19 ± 0.01 i	0.17 ± 0.01 gh	0.29 ± 0.02 ef
		DAP	45 ± 1.63 d	64.1 ± 1.59 hi	30.5 ± 1.46 b	1.7 ± 0.26 e–g	8.3 ± 0.25 a–d	0.23 ± 0.01 hi	0.18 ± 0.02 f–h	0.34 ± 0.01 de
	LAOM	RP	52.67 ± 1.2 a–c	72.3 ± 1.5 f–h	25 ± 0.58 c	2.64 ± 0.21 d	8.46 ± 0.16 0	0.35 ± 0.01 c–f	0.27 ± 0.02 b–e	0.45 ± 0.02 a–c
		SSP	51 ± 0.58 bc	74 ± 1.53 e–ch	25 ± 0.58 c	2.65 ± 0.26 d	8.88 ± 0.46 a–d	0.3 ± 0.02 fg	0.25 ± 0.01 d–f	0.43 ± 0.02 bc
		DAP	52.3 ± 1.86 a–c	81.3 ± 1.8 c–f	32.3 ± 0.8 ab	3.1 ± 0.31 b–d	9.11 ± 0.17 a–d	0.35 ± 0.01 c–f	0.28 ± 0.01 b–e	0.44 ± 0.01 bc
	AOM	RP	56.33 ± 1 ab	78.3 ± 1.7 d–cf	30.7 ± 1.53 b	2.5 ± 0.31 de	7.88 ± 0.27 cd	0.37 ± 0.01 b–f	0.3 ± 0.01 a–e	0.46 ± 0.01 a–c
		SSP	52.7 ± 1.86 a–c	77.7 ± 2.1 d–g	25.7 ± 1.17 c	2.69 ± 0.24 d	8.64 ± 0.19 a–d	0.34 ± 0.01 d–g	0.27 ± 0.01 b–e	0.45 ± 0.01 a–c
		DAP	54 ± 1.53 a–c	85.3 ± 1.8 a–d	33.3 ± 0.8 ab	3.12 ± 0.1 b–d	8.48 ± 0.43 a–d	0.37 ± 0.02 b–f	0.3 ± 0.01 a–e	0.46 ± 0.01 a–c
PSB (+)	CK	RP	52 ± 1.08 a–c	68 ± 1.48 gh	25.8 ± 1.21 c	2.33 ± 0.2 d–f	8.07 ± 0.61 b–d	0.33 ± 0 e–g	0.26 ± 0.02 de	0.39 ± 0.01 cd
		SSP	49.3 ± 0.73 cd	57.17 ± 1.33 i	23.8 ± 1.81 c	2.34 ± 0.25 d–f	8.47 ± 0.42 a–d	0.28 ± 0.02 gh	0.24 ± 0.01 e–g	0.4 ± 0.01 cd
		DAP	52.7 ± 0.73 a–c	71.3 ± 1.7 f–h	31.7 ± 1.18 b	2.83 ± 0.23 cd	8.39 ± 0.51 a–d	0.3 ± 0.02 fg	0.27 ± 0.01 c–e	0.42 ± 0.02 b–d
	LAOM	RP	55.7 ± 0.88 ab	87.3 ± 1.4 a–cd	32.1 ± 0.6 ab	3.67 ± 0.17 a–c	8.57 ± 0.55 0	0.42 ± 0.02 a–c	0.32 ± 0.01 a–d	0.46 ± 0.02 a–c
		SSP	53.3 ± 0.88 a–c	82.7 ± 2 b–ce	32.3 ± 0.6 ab	3.67 ± 0.35 a–c	8.93 ± 0.69 a–d	0.39 ± 0.02 a–e	0.29 ± 0.02 b–e	0.46 ± 0.01 a–c
		DAP	56.3 ± 0.88 ab	92.3 ± 2.7 ab	33.8 ± 0.9 ab	3.68 ± 0.3 a–c	9.28 ± 0.52 a–d	0.43 ± 0.01 ab	0.34 ± 0.01 a–c	0.5 ± 0.02 ab
	AOM	RP	57.67 ± 0.88 a	90.7 ± 1.7 a–c	34 ± 0.58 ab	3.9 ± 0.06 ab	10.08 ± 0.2 ab	0.45 ± 0.01 a	0.34 ± 0.01 ab	0.48 ± 0.02 a–c
		SSP	54 ± 0.58 a–c	83.7 ± 0.8 b–e	32.5 ± 0.8 ab	3.69 ± 0.06 a–c	9.82 ± 0.23 a–c	0.41 ± 0.01 a–d	0.31 ± 0.01 a–e	0.48 ± 0.02 a–c
		DAP	57 ± 1.53 a	95.3 ± 1.77 a	36.3 ± 0.33 a	4.11 ± 0.08 a	10.22 ± 0.5 a	0.45 ± 0.01 a	0.36 ± 0.01 a	0.53 ± 0.01 a

**Table 3 plants-12-03072-t003:** Effect of LAOM and AOM with different P sources along PSB on maize seed quality-related parameters in grains and weight of thousand grains (g): crude fat (%), crude fiber (%), crude protein (%), and ash contents (%). Different letters along with figures indicate the significant differences (*p* < 0.05) of PSB and acidified organic matter to P fertilizers.

Bacteria	O.M	P Source	Crude Fat (%)	Crude Fiber (%)	Crude Protein (%)	Crude Protein (%)	KGW (g)
PSB (−)	CK	RP	0.49 ± 0.03 j	1 ± 0.03 k	6.14 ± 0.12 h	0.74 ± 0.02 h	164.8 ± 4.3 g
		SSP	0.61 ± 0.02 ij	1.14 ± 0.03 jk	6.68 ± 0.07 gh	0.8 ± 0.02 h	189 ± 3.14 g
		DAP	0.75 ± 0.03 gh	1.61 ± 0.04 hi	7.83 ± 0.11 d–f	1.19 ± 0.03 e–g	226 ± 3.53 f
	LAOM	RP	0.72 ± 0.02 hi	2.02 ± 0.05 c–f	7.78 ± 0.1 d–f	1.28 ± 0.02 d–f	338 ± 5.7 bc
		SSP	0.71 ± 0.02 hi	1.9 ± 0.04 e–g	7.75 ± 0.1 ef	1.25 ± 0.03 e–g	325 ± 2.9 b–d
		DAP	0.91 ± 0.02 d–f	2.2 ± 0.09 a–e	8.71 ± 0.23 bc	1.48 ± 0.04 b–d	304.7 ± 5.2 de
	AOM	RP	0.9 ± 0.02 b–d	2.06 ± 0.07 b–f	8.18 ± 0.09 c–e	1.38 ± 0.02 c–e	283.7 ± 2.9 e
		SSP	0.92 ± 0.02 c–e	1.95 ± 0.02 d–g	7.75 ± 0.08 ef	1.3 ± 0.02 d–f	335 ± 2.9 bc
		DAP	0.9 ± 0.01 b–d	2.26 ± 0.05 a–c	8.8 ± 0.26 bc	1.54 ± 0.06 a–c	315 ± 2.9 cd
PSB (+)	CK	RP	0.8 ± 0.01 e–h	1.69 ± 0.05 g–i	7.62 ± 0.08 ef	1.13 ± 0.02 fg	219.1 ± 2.9 f
		SSP	0.79 ± 0.02 f–h	1.41 ± 0.04 ij	7.34 ± 0.14 fg	1.06 ± 0.03 g	225.7 ± 4.1 f
		DAP	0.8 ± 0.01 d–g	1.84 ± 0.04 f–h	7.92 ± 0.1 d–f	1.36 ± 0.02 c–e	317.3 ± 7.7 cd
	LAOM	RP	0.92 ± 0.02 c–e	2.24 ± 0.09 a–d	8.72 ± 0.2 bc	1.37 ± 0.04 c–e	345.3 ± 6.9 b
		SSP	0.9 ± 0.03 d–f	2.15 ± 0.09 a–e	8.33 ± 0.2 c–e	1.37 ± 0.03 c–e	345 ± 2.89 b
		DAP	1.04 ± 0.05 a–c	2.37 ± 0.07 a	9.37 ± 0.2 ab	1.62 ± 0.04 ab	382.7 ± 12.7 a
	AOM	RP	1.06 ± 0.02 ab	2.33 ± 0.04 ab	9.27 ± 0.09 ab	1.52 ± 0.02 a–c	338.3 ± 4.4 bc
		SSP	1.05 ± 0.04 ab	2.22 ± 0.05 a–e	8.5 ± 0.15 cd	1.52 ± 0.07 a–c	350 ± 2.89 b
		DAP	1.17 ± 0.03 a	2.43 ± 0.07 a	9.53 ± 0.1 a	1.7 ± 0.03 a	395 ± 4.94 a

## Data Availability

The original data presented in the current study is available upon request to the corresponding author.

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
