# Peer review of "Bio-Organically Acidified Product-Mediated Improvements in Phosphorus Fertilizer Utilization, Uptake and Yielding of Zea mays in Calcareous Soil"

_plants, 2023, doi:10.3390/plants12173072_

Round 1

Reviewer 1 Report

In this manuscript, the authors attempt to demonstrate how phosphorus use efficiency can be improved in Zea mays by using an acidified product. However, there are major concerns regarding this work. The authors did not adequately justify the use of maize or the bacterial strains used in the study. Additionally, while the authors compare growth rates, they do not provide any photographs to support their results.

To improve this manuscript, the authors should introduce and justify their choices to build a strong narrative that supports their results. Currently, the manuscript mostly consists of descriptions without any plant images to support the conclusions. Furthermore, some conclusions are not supported by the study's results, such as the statement that "physiological attributes, nutrient uptake, and seed quality parameters of maize were improved." There is no study or results about seed quality in this paper, so this conclusion lacks support.

Author Response

Thank you for the nice comment. We believe that the quality of the manuscript is much improved after incorporating the suggested comments by honorable reviewers. The detail response of the comments is as under.

1-In this manuscript, the authors attempt to demonstrate how phosphorus use efficiency can be improved in Zea mays by using an acidified product. However, there are major concerns regarding this work. The authors did not adequately justify the use of maize, or the bacterial strains used in the study. Additionally, while the authors compare growth rates, they do not provide any photographs to support their results.

Response: Thanks for your comment and pointing out about plant growth figure addition. We have briefly discussed information about bacterial strain in methodology section heading 2.2 which were used to process the manure to acidified product, and similarly mentioned the required data about maize in heading 2.4. Besides, the role of both bacteria and maize we have discussed very clearly in discussion part (4) even in all sections of article. Secondly, we have provided the photograph of maize growth and named with Figure 1 as per your suggestion.

2- To improve this manuscript, the authors should introduce and justify their choices to build a strong narrative that supports their results. Currently, the manuscript mostly consists of descriptions without any plant images to support the conclusions. Furthermore, some conclusions are not supported by the study's results, such as the statement that "physiological attributes, nutrient uptake, and seed quality parameters of maize were improved." There is no study or results about seed quality in this paper, so this conclusion lacks support.

Response: Thank you for your comment. As we already have added plants growth image named with Figure 1 as per your kind suggestion. Further, in this study we proposed a new idea of soil pH manipulation by adding bio-augmented organic manure to improve efficient fertilizer use and soil P use efficiency. Our study results indicated that the using this bio-organic product not only improve soil p availability and efficient utilization of rock phosphate but also improve the maize growth and yield as well. Keeping the data (about physiological attributes, seed quality parameters, and nutrient (P) uptake) in tables 2 and 3, and figure 2 respectively mentioned, which clearly justify our study conclusion as well as verify overall objective of current study. For seed quality and yield related parameter we have added both together in Table 3 and interpret the results in heading 3.3.

Reviewer 2 Report

Dear Authors,

Congratulation for the good work. Some notes

1. Please check one more time the style of references.

2. I detected 26% plagiarism. Please decrease it

3. Change the section Discussion to 4. Discussion

The English language is fine.

Author Response

Thank you for the valuable comment. We believe that the quality of the manuscript is much improved after incorporating the suggested comments by honorable reviewers. The detailed response to the comments is as below.

1-Please check one more time the style of references.

Response: Thanks for your suggestion. We have clearly checked reference section.

  1. I detected 26% plagiarism. Please decrease it

Response: Thanks for pointing it out. We have reduced the similarity index as per your kind suggestion.

  1. Change the section Discussion to 4. Discussion

Response: Thanks for your kind suggestion. I have changed it.

Reviewer 3 Report

Dear Authors, 

I find it an interesting article.  

I think that the introduction is good, with sufficient and quite adequate citations, in addition, it presents a hypothesis and a well-defined objective.  

In the Material and Methods section there are some aspects that should be improved: 

-Add the method with which the organic matter was calculated.

-The method for calculating total P was added.

-The method with which the assimilable K was calculated has been added.

-The method used to calculate total N was added.

When commenting on the "recommended" N and K dosage (181:131kg ha-1), the bibliographic citation should be included. 

Figure 1: its legend must be self-explanatory. It should explain each of the four graphs (A) (B) (C) and (D). And refer in the text to Figure 1A or Figure 1B..... It should also add the meaning of the acronym CK. 

Figure 2: its legend must be self-explanatory. It should add to each graph (A) and (B), and refer in the text to Figure 2A or 2B. It should also add the meaning of the acronym CK.

Correct the legends of figures 3 and 4, they are missing the dots (Figure 3.) and the sentence endpoints.

In the section "Relationship between maize growth...", you should put parentheses when referring to Figure 3 or Figure 4 in the text, there are several errors of this type in this section. Example: ...between maize growth attributes (Figure3). 

The discussion section seems correct to me as well as the conclusions.

Author Response

I find it an interesting article.  

I think that the introduction is good, with sufficient and quite adequate citations, in addition, it presents a hypothesis and a well-defined objective.

Response: Thank you for the nice comment. The manuscript has been revised as per honorable suggestion. The detailed response to the comments is as below.

In the Material and Methods section there are some aspects that should be improved: 

-Add the method with which the organic matter was calculated.

-The method for calculating total P was added.

-The method with which the assimilable K was calculated has been added.

-The method used to calculate total N was added.

When commenting on the "recommended" N and K dosage (181:131kg ha-1), the bibliographic citation should be included. 

Response: Thanks for your useful suggestions. we have incorporated methods of N, P, K calculations and added citation about recommended dose rate of N, P, K as you mentioned (heading 2.3). Also, we have revised the methodology section as required.

Figure 1: its legend must be self-explanatory. It should explain each of the four graphs (A) (B) (C) and (D). And refer in the text to Figure 1A or Figure 1B..... It should also add the meaning of the acronym CK. 

Response: Thanks for your useful suggestion. We have revised all Figure numbering and revised figure 1 with new number figure 2. So, we revised legends of figure 2 according to your suggestion. CK mentioned the control treatment, we have also explained in methodology section as well in figures captions. Further, we revised the all-figures titles and captions in this article.

Figure 2: its legend must be self-explanatory. It should add to each graph (A) and (B), and refer in the text to Figure 2A or 2B. It should also add the meaning of the acronym CK.

Response: Thanks for nice suggestion. We have revised all figure numbering and revised figure 2 with new number figure 3. So, we have revised figure 3 and all graphs’ legends are self-explanatory and mention in figure caption as CK mention the control treatment accordingly.

Correct the legends of figures 3 and 4, they are missing the dots (Figure 3.) and the sentence endpoints.

Response: Thanks for pointing it out. We have revised Figure 3,4 numbering and gave new number with Figure 4 and 5. Also, we have cross checked and added the missing dots at the end of sentences where required.

In the section "Relationship between maize growth...", you should put parentheses when referring to Figure 3 or Figure 4 in the text, there are several errors of this type in this section. Example: ...between maize growth attributes (Figure3). 

Response: Thanks for pointing it out. We have revised the figure number 3 and 4 with new figure number 4 and 5. We have added the figures numbers in text of heading 3.5. as you suggested. Also, I have revised in whole article where required.

The discussion section seems correct to me as well as the conclusions.

Response: Thanks for your supportive comment.

Reviewer 4 Report

The article is very interesting, but needs to be greatly improved before publishing:

1. in the title, replace the word production with yielding

2. write the key words in lowercase letters

3. introduction correctly written, but needs to be added to clarify the research hypothesis

4. conclusions - add recommendations for agricultural production

5. in formulas add, insert units

6. methodology needs to be refined. How many years, series of studies, how the pots were watered, volume of the pot, how many plants in the pot,

7. Discussion - in pierereparagraph necessarily add a few sentences about row application, starter phosphorus in corn cultivation. For this purpose, it is necessary to quote some articles.

Author Response

The article is very interesting, but needs to be greatly improved before publishing:

Response: Thank you for the nice comment. We believe that the quality of the manuscript is much improved after incorporating the suggested comments by honorable reviewers. The detailed response to the comments is as below.

1-in the title, replace the word production with yielding

Response: Thanks for nice suggestion. We have incorporated the required changes.

  1. write the key words in lowercase letters

Response: Thank you for suggestion. We have incorporated the mentioned changes.

  1. introduction correctly written, but needs to be added to clarify the research hypothesis

Response: Thanks for your suggestion. We have added the study hypothesis in the last paragraph of the introduction part.

  1. conclusions - add recommendations for agricultural production

Response: Thanks for your suggestion. We have considered and recommended future agriculture production.

  1. in formulas add, insert units

Response: Thanks for pointing it out. We have added in heading 2.5.

  1. methodology needs to be refined. How many years, series of studies, how the pots were watered, volume of the pot, how many plants in the pot,

Response: Thanks for your comment. We have mentioned in 2.4 second last paragraph.

  1. Discussion - in pierereparagraph necessarily add a few sentences about row application, starter phosphorus in corn cultivation. For this purpose, it is necessary to quote some articles.

Response: Thanks for your suggestion. The discussion section has been improved as per suggestion.

Reviewer 5 Report

General comments 

The manuscript describes an interesting and important research topic. Instead of its many strength, I now highlight the the parts that need improvement.

It would have been easier to write the review if the lines were numbered, and perhaps more followable.

In the manuscript, many measurement results are published, but their (statistical) evaluation is incomplete, the results are often lost in the details.

The formal requirements are sometimes not met in the text, fonts of different types and sizes appear randomly (especially in reference numbering). The formatting of the large tables should also be improved, in their current state they are difficult to read and differ from the font of the text. The use of the expressions “however, moreover, overall, since, similarly, interestingly” is imprecise and often does not correspond to their meaning.

Detailed comments

Abstract

Like with general comments.

Introduction

The aim of the research was “our research study hypothesized that the use of acidulated organic manure and microbes in combination with synthetic P-fertilizers may increase the plant biomass, nutritional quality, and yield of maize through improved P utilization”. The literature review provides a suitable background for this.

Materials and methods

2.1. The elemental composition of acidulated organic manure (AOM) was reported. Why was the composition of LAOM not announced? And the pH values are missing for both. Why?

2.3. The soil type (according to one of the chosen taxonomy), water holding capacity and carbonate contents are essential for the correct interpretation of the experimental results. What method was used to measure the available K? How were the amounts of fertilizer given per hectare converted to pot amounts (8 kg soil)? In what form were N and K fertilizers used?

Probably the statistical analysis is the Chapter 2.6.

Results

The Results chapter is quite long, but it contains several sections that would be better placed either in the introduction or in the Discussion. For example, the description of treatments and their abbreviations and names belong in the Materials and methods chapter.

The Tables 1-3 should be reformatted to make them more readable. Or transfer it to in the Supplement and instead of them present the results of the two or three-factor analysis of variance. This is missing, even though the three-factor ANOVA would indicate whether there was a significant effect of bacterial inoculation, organic supplements or the form of fertilizer.

Figure 2 contains information about bacterial count. But previously there was no sign of this measurement. The description a culturing conditions is missing from the 2. Chapter. Also, in the Fig 2 there is SOC measurement, but their values quite low (2-8 mg/kg). Is it possible?

Instead of Figure 3, I suggest the “traditional” method for reporting result: in a table with numbers.

Figure 4 (PCA analysis) needs more detailed explanation. And a more detailed list of abbreviations. Some abbreviations missing (e. g. for MBC).

 Discussion

The evaluation of the results was partially done in the previous chapter (3. Results), but it should be in this chapter. However, this is missing, and some topics are mentioned that should be in the introduction (e. g. rhizosphere pH, action mechanisms of P solubilizing bacteria, leaves stromata, photosynthetic pigments).  I recommend rewriting this chapter, better emphasizing and evaluating the research results.

Best regards, 

Author Response

General comments 

The manuscript describes an interesting and important research topic. Instead of its many strength, I now highlight the parts that need improvement.

1-It would have been easier to write the review if the lines were numbered, and perhaps more followable.

Response: We are thankful to you and accordingly we have mentioned the line numbers throughout the manuscript for your ease for further revision.

2-In the manuscript, many measurement results are published, but their (statistical) evaluation is incomplete, the results are often lost in the details.

Response: Respected reviewer, we critically applied the statistical analysis on recorded data as mentioned in tables and figures, where you can easily note the statistically significant differences of different treatments using p<0.05 significance level. Alternatively, in figure 2D the analysed data of PUE (P use efficiency %) was calculated from the mean values of treatments that’s why it has not the replications data, so we mentioned their results at labels of graph bars accordingly. Secondly results section is briefly described and arranged in such a manner to understand easily, as concise, and critical changes with respective controls were mentioned considering the length of article.

3-The formal requirements are sometimes not met in the text, fonts of different types and sizes appear randomly (especially in reference numbering). The formatting of the large tables should also be improved, in their current state they are difficult to read and differ from the font of the text. The use of the expressions “however, moreover, overall, since, similarly, interestingly” is imprecise and often does not correspond to their meaning.

Response: Thanks for your kind comment. As per your suggestions we have revised fonts of different types and sizes in whole article. Further, the format of tables and use of expression word also fixed in updated article.

Detailed comments

Detailed comments

Abstract

Like with general comments.

Introduction

The aim of the research was “our research study hypothesized that the use of acidulated organic manure and microbes in combination with synthetic P-fertilizers may increase the plant biomass, nutritional quality, and yield of maize through improved P utilization”. The literature review provides a suitable background for this.

Materials and methods

2.1. The elemental composition of acidulated organic manure (AOM) was reported. Why was the composition of LAOM not announced? And the pH values are missing for both. Why?

Response: We are delighted that reviewer critically reviewed and the pH value of acidified organic amendment have been mentioned in revised manuscript, similarly the chemical composition of LAOM was not mentioned as LAOM does not showed significant results as compared to AOM, while this LAOM was prepared through dilution of AOM at 1:2.5 (w/v) and good quality water was used so significant difference of their composition was not monitored rather that high moisture as slurry like structure. So, the most efficient product’s composition was announced for better understanding and less confusion to readers.

2.3. The soil type (according to one of the chosen taxonomy), water holding capacity and carbonate contents are essential for the correct interpretation of the experimental results. What method was used to measure the available K? How were the amounts of fertilizer given per hectare converted to pot amounts (8 kg soil)? In what form were N and K fertilizers used?

Response: Thanks for your useful comments. For present study, I have added the suggested information’s in the section 2.3. Secondly, the standard method of Estefan, G. Methods of soil, plant, and water analysis: a manual for the West Asia and North Africa region. 2013. Was used for available K method, similarly the prescribed method also mentioned in updated MS version. For Nitrogen (N) application, we used urea (46% N) and for potassium (K) we used muriate of potash (MOP 60% K2O). Lastly, Amount of required fertilizer for 8 kg soil was calculated using the standard proposed formula for fertilizers calculation based on soil weight, as total weight of one hectare soil is 2×106 kg and similarly the recommended amount of fertilizer was calculated for 8 kg soil using unit method of calculation.

Probably the statistical analysis is the Chapter 2.6.

Response: Yes, Respected reviewer and changes have been mentioned.

Results

The Results chapter is quite long, but it contains several sections that would be better placed either in the introduction or in the Discussion. For example, the description of treatments and their abbreviations and names belong in the Materials and methods chapter.

Response: Agree with you Dear reviewer, we have critically reviewed the results section and more briefly we justify our results and accordingly the connecting general sentences have been removed from this section, while the first paragraph of results contains concise but mandatory description for the readers easiness and better understandings.

The Tables 1-3 should be reformatted to make them more readable. Or transfer it to in the Supplement and instead of them present the results of the two or three-factor analysis of variance. This is missing, even though the three-factor ANOVA would indicate whether there was a significant effect of bacterial inoculation, organic supplements or the form of fertilizer.

Response: Yes, Dear Professor, we have reformatted these tables to make more readable as through zooming in readers could easily interpret, similarly we are giving these tables in main MS as more easily accessible and understandable for our readers. Secondly, according to your supportive suggestion for three factor ANOVA that we are giving as a supplementary data where there is a significant effect of bacterial inoculation, organic amendments, and the forms of fertilizers could be evaluated.

Figure 2 contains information about bacterial count. But previously there was no sign of this measurement. The description a culturing conditions is missing from the 2. Chapter. Also, in the Fig 2 there is SOC measurement, but their values quite low (2-8 mg/kg). Is it possible?

Response: Thanks for pointing it out. We have added in heading 2.2, the bacterial count was analysed by using serial dilution plating technique.  Similarly, SOC overlapped and now we have corrected it in revised version (mg/g).

Instead of Figure 3, I suggest the “traditional” method for reporting result: in a table with numbers.

Response: Respected reviewer, we preferably tried to present our data in tabular form and most of the data have been already presented whereas the microbial count and SOC are core findings of the proposed study and are crucial for reader interest so we have decided to kept as it in the form of figure.

Figure 4 (PCA analysis) needs more detailed explanation. And a more detailed list of abbreviations. Some abbreviations missing (e. g. for MBC).

Response: Thanks for pointing it out. We have revised the Figure 4 caption as per your kind suggestion, and we have added the missing abbreviations.

Discussion

The evaluation of the results was partially done in the previous chapter (3. Results), but it should be in this chapter. However, this is missing, and some topics are mentioned that should be in the introduction (e. g. rhizosphere pH, action mechanisms of P solubilizing bacteria, leaves stromata, photosynthetic pigments).  I recommend rewriting this chapter, better emphasizing and evaluating the research results.

Response: Thanks for your useful comment. we have revised and improved the results section as well as discussion section accordingly.

Round 2

Reviewer 1 Report

Thanks for improving the manuscript following my recommendations.

Reviewer 4 Report

Thank you to the authors for taking in to account my comments and suggestions.

The article recommends for publication

Reviewer 5 Report

The author significantly improved the manuscript, I suggest the acceptance.

Best regards